# Does Internet Use Promote Subjective Well-Being? Evidence from the Different Age Groups Based on CGSS 2017 Data

**DOI:** 10.3390/ijerph20042897

**Published:** 2023-02-07

**Authors:** Yurong Yan, Yuying Deng, Juan-José Igartua, Xiagang Song

**Affiliations:** 1School of Journalism and Communication, Northwest University of Political Science and Law, Xi’an 710122, China; 2Department of Sociology and Communication, Faculty of Social Sciences, University of Salamanca, 37007 Salamanca, Spain; 3School of Law, Shihezi University, Shihezi 832000, China

**Keywords:** internet use, subjective well-being, use frequency, online relationship size, internet proficiency

## Abstract

Mobile Internet technology has developed so rapidly that the Internet has become indispensable in everyday life. There is a continuous debate about the relationship between internet use and subjective well-being. In contrast to observing whether one has Internet access, this paper focuses on three dimensions of Internet usage: frequency of use, online relationship size, and Internet proficiency. Based on the Chinese nationwide data collected in 2017, the results of the ordinary least squares regression model demonstrate that Internet use has a significant positive association with subjective well-being. In addition, this study also discovers that the effect of Internet use on the subjective well-being of individuals of different ages is heterogeneous; middle-aged individuals benefit from more frequent Internet use and larger-scale networks; the young and older adults benefit from organizing communication in groups. The results of this study can provide targeted suggestions for improving the subjective well-being of different age groups in Internet use.

## 1. Introduction

Internet technology has developed so rapidly that the Internet has become indispensable in everyday life. Individuals spend an average of 7 h on the Internet every day, 145 min of which are spent on social media [1]. As a transformative force that promotes social restructuring [2], does the Internet also contribute to subjective well-being? The Internet provides a wealth of information resources and a wide range of communication channels that can help users to acquire different kinds of information, strengthen and broaden their relationships, exchange personal opinions, and generate social capital [3,4,5]. At the same time, Internet use is thought to be related to online harassment, poor sleep quality, low self-esteem, and poor body image, all of which are associated with higher depression scores [6]. Excessive attachment to the use of the Internet may compromise real-world interactions that provide social support [7] and cause individuals to neglect their responsibilities at work and in the family home [8].

Subjective well-being in the political, economic, and natural environment is a pressing concern in China [9]. The Chinese government treats happiness, security, and sustainability as elements of its social development agenda [10]. Subjective well-being is linked to a variety of positive outcomes, including improved health and longevity [11,12]. Innovations that improve well-being should be valued both in practice and in empirical research [13].

Many studies have focused on the influence of the Internet on subjective well-being, but the relevant associations, be they positive or negative, are unclear [14]. The meta-analyses that have been published to date show that there is a statistically significant negative correlation between social media use and well-being [15,16]. The specific results depend on how Internet use and subjective well-being are defined and operationalized [17].

Furthermore, much of the relevant research has been conducted in affluent countries [18]. For example, Greyling studied Gauteng province, the economic hub of South Africa, and only focused on Internet access [19]. The manner in which individuals use the Internet and the identities of those who do should not be neglected [20]. Moreover, studies of the relationship between Internet use and subjective well-being heavily rely on data from adolescents and young adults [14]. However, the Internet has gradually spread among middle-aged and older people, which means that researchers should recruit a wider range of participants across more varied age brackets. Paez and colleagues analyzed the effects of Internet use on subjective well-being by studying nationally representative samples from 18 nations, some of which were affluent and some of which were not. However, they did not find differences between age groups [21].

As of August 2022, 1.051 billion individuals use the Internet in China [22]. The purpose of this study is to explore the association between Internet use behaviors and personal subjective well-being in the Chinese context. Unlike other studies that focus on Internet access, this paper considers subjective well-being in relation to the frequency of Internet use, online relationship size, and Internet proficiency. It also investigates use patterns in different age groups and their influence on subjective well-being.

## 2. Literature Review

### 2.1. Subjective Well-Being

The literature on subjective well-being emerged out of American psychology in the 1950s [23]. Its concern is with the views that individuals hold about their well-being [7,24]. Well-being can be described as how well people are doing in life, taking into account social, health, material, and subjective components of well-being [25]. Subjective well-being is a holistic assessment of an individual’s life quality that is based on internal criteria, and it is an important and comprehensive psychological indicator of life quality [26]. Subjective well-being is significant for health [27], intimate relationships [28], and career success [29]. Diener and colleagues argued that subjective well-being has three components: pleasant affect, unpleasant affect, and life satisfaction [26]. Empirical studies have also investigated the different aspects of subjective well-being. Li and Zhou included self-confidence in their measurements [30]. In addition to happiness and life satisfaction, other researchers accounted for a general indicator of subjective well-being [31]. Previous research has shown that subjective well-being is significantly correlated with sociodemographic characteristics, including sex, income, age, and educational attainment [31,32,33]. It has also been shown that married individuals are happier and more satisfied with their lives than singles [34].

### 2.2. Internet Use Behaviour and Subjective Well-Being

There is an ongoing debate about the relationship between Internet use and well-being. Previous studies have revealed a significant correlation between the two. The Internet is thought to influence subjective well-being by changing how time is allocated to various activities, creating new activities, facilitating access to information, and by enriching communication [7]. Pera and colleagues revealed that older adults experience superior well-being when they share photos on the Internet, which enables them to connect to others, irrespective of their location or their personal situation [35]. Similarly, Tuenti has become a suitable platform for the establishment, consolidation, and expansion of social relationships among Spanish adolescents, and socializing on Tuenti significantly improves adolescents’ perceptions of their own well-being [36].

Conversely, more exposure to the Internet is likely to be associated with lower levels of subjective well-being. According to a comparative study of American and Finnish teenagers, those who are exposed to hate speech online have lower levels of happiness [37]. In a meta-analysis, Huang found that Internet use has a very weak negative effect on subjective well-being [38]. Although scholars have sought to uncover the association between Internet exposure and subjective well-being in general, most results reflect the use of European or North American data. Notably, the claim that excessive Internet use decreases subjective well-being was found to be unsupported in a study that was based on data from 18 countries [21].

One empirical study that drew on a Chinese sample confirmed that the Internet contributes to multi-dimensional well-being, including physical, mental, and social health [39]. More recently, Jiang and Chen studied a sample of older adults from the 2017 Chinese General Social Survey (CGSS) and found that Internet usage was positively associated with subjective well-being [40]. Accordingly, the first research question of this paper is as follows:

RQ1: is Internet use associated with subjective well-being in a representative all-ages sample from the Chinese population?

Given the rapid development of online platforms, Internet use behaviors cannot be measured by reference to intensity alone [41]. At the same time, the relationship between Internet use and mental health merits further exploration. The Internet is ubiquitous, and its uses considerably vary. Different usage behaviors can lead to different results [42]. For example, the authors of one study found that a one-hour increase in social media use leads to a decrease in user happiness of between 0.17 and 0.22 [43]. Long-term engagement on the Internet entails spending much time on communication and the maintenance of social networks. Adolescents fear missing information, which may generate anxiety and thus harm their subjective well-being [44]. However, a Chinese study suggests that using the Internet more frequently to study, work, and socialize has a positive impact on subjective well-being [45]. Wheatley and Buglass revealed that web usage has increased across age groups in England, and life satisfaction is higher among those who use social media for an hour or less each day. However, it is lower among individuals who use the Internet for four or more hours [46]. Therefore, we formulated the following hypothesis:

**H1.** *More frequent internet use is associated with higher subjective well-being*.

Beyond length or frequency of usage, the capacity of the Internet to facilitate interpersonal connections is enhanced by the relational character of the network. Numerous studies have suggested that online communication results in more interactive relationship networks and thus further improves subjective well-being. In a study of 527 male adolescents in Northern Ireland, those with more online friends were found to score higher for subjective well-being [47]. Similarly, Ishii, after conducting a survey of Japanese social media users, argued that the number of friends on a social networking service can influence subjective well-being [48]. According to Nabi and colleagues, Internet users evaluate their social support availability based on their number of friends, regardless of the precise nature of those connections or the messages received [49]. In other words, possessing a large number of friends on the Internet is considered to be beneficial for individuals. However, Kim and Lee demonstrated that the number of Facebook friends has an inverse U-shaped relationship with subjective well-being [50], which emphasizes the importance of devoting sufficient time to the development of connections with real-world friends. Having considered these results, we hypothesized as follows:

**H2.** *Individuals with a larger online relationship size have higher levels of subjective well-being*.

The digital economy and society emphasize individuals’ advanced skills and development. The ability to create content shared with others online is considered a higher-level skill relative to basic operational ones [51]. A person’s proficiency with using the Internet denotes that he or she can select, evaluate, and act on various contacts, which also relates to a central network position and has been found to be associated with subjective well-being. According to Fowler and Christakis, there is an association between online relationship centrality and happiness [52], that is, individuals at the core of a network are more likely to be happy, while individuals on the periphery of a network are more likely to be unhappy. Also, a similar conclusion is reached in a study of Chinese senior undergraduates [53]. The Internet is creating conditions that enable individuals to become influential in their relationship networks. Virtual communities, which may revolve around information, relationships, fantasies, or transactions, provide specific benefits to their members [54]. Activities and interactions in such communities can induce loyalty [55] and happiness [56] among members. The administrators of group chats are expected to be responsible for the management of the groups as well as for maintaining order [57,58]. They can send messages to subsets of users or the entire group at once [59]. They may experience a higher level of subjective well-being as a result of being at the center of these dispersed groups of individuals who have similar socialization needs. Accordingly, we formulated the following hypothesis:

**H3.** *Individuals with higher levels of Internet use proficiency have higher levels of subjective well-being*.

Several scholars have focused on the relationship between Internet use behaviors and subjective well-being in different age groups. Some researchers have concluded that the daily overuse of media by children might cause beneficial face-to-face communication to be forgone [60,61]. Digital media also provide children with freedom and autonomy and enhance their well-being [62]. A study examined how the intensity of Facebook use could reduce inequalities in well-being among undergraduates at a U.S. university [63]. In a study of the use of pro-eating disorder websites by young individuals from four countries, researchers argued that those who are exposed to such websites experience less happiness and advertise harmful dieting to individuals without histories of eating disorders [64]. Others have identified a positive association between Internet use and the subjective well-being of older individuals. Specifically, Internet access may enable older individuals to maintain close intergenerational relationships [30]. The members of different age groups have different foci when they use the Internet, which, in turn, leads to different associations between Internet use behaviors and subjective well-being. Therefore, we formulated an additional research question:

RQ2: does Internet use have different effects on different age groups?

## 3. Method

### 3.1. Data

This study draws on data from the 2017 Chinese General Social Survey (CGSS), which is the most recent data from the CGSS. It is the earliest national, comprehensive, and continuous academic survey project in China and generates data that is representative of contemporary Chinese society [65]. The number of people using the Internet in China has grown by around 300 million people since 2017 [22,66]. Analysis of this representative sample of data can also draw further attention to the impact of current Internet use on the subjective well-being of people of different ages. The 2017 CGSS data includes modules such as “network society” and “social network”, which facilitate research on Internet usage behaviors. Once the missing and the invalid variables had been removed, a total of 2079 valid samples were obtained.

### 3.2. Measures

#### 3.2.1. Outcome Variable

The outcome variable is “subjective well-being” (SWB). Following Wang and Liang, the dependent variable was measured by asking respondents whether they thought that their lives were happy in general [67]. The single question on the level of SWB is reliable and effective [68]. The values that were allocated to answers range from 1 (“very unhappy”) to 5 (“very happy”) (M = 3.90, SD = 0.79). We also created the variable “life satisfaction” to test for robustness. It is based on the following question from the CGSS: “In general, are you satisfied with your living conditions?” [69]. The values range from 1 (“very dissatisfied”) to 7 (“very satisfied”). The higher the values, the higher the level of life satisfaction (M = 4.88, SD = 1.01).

#### 3.2.2. Predictor Variable

We considered three predictor variables. First, we measured the frequency of Internet usage by using the question “In the past year, did you surf the Internet often in your spare time?”. We assigned values to the answers that range from 1 to 5 (1 = “never”; 2 = “several times a year or less”; 3 = “several times a month”; 4 = “several times a week”; 5 = “every day”) (M = 4.09, SD = 1.08). Second, we determined the online relationship size by drawing on the respondents’ answers to the question about the number of individuals with whom they would connect via the Internet every day (M = 3.12, SD = 1.20). Third, we measured Internet proficiency by examining the respondents’ answers to the question about organizing virtual communities, either organizing WeChat or QQ groups. A value was assigned to the number of virtual communities that they organized (M = 0.98, SD = 3.05).

#### 3.2.3. Control Variables

The demographic variables that we used as controls include sex (0 = “female”, 1 = “male”), age (in years), marital status (0 = “single, divorced, or widowed”,1 = “married”), education level (0 = “uneducated”, 6 = “private school or elementary school”, 9 = “junior high school”, 12 = “senior high school or technical school”, 16 = “college diploma”, 19 = “bachelor’s degree or higher”), and annual income (annual income from the preceding year transformed by natural logarithm). The literature has shown that there is a U-shaped relationship between age and life satisfaction [70]. Thus, we included the squared age variable to control for the non-linear effects of age.

### 3.3. Data Analysis

We used Stata 17 to analyze the data. Means, standard deviations (SD), frequencies, and percentages were calculated to describe the demographics of the sample and to capture other information about its characteristics, Internet usage, and subjective well-being. To explain how Internet usage affects subjective well-being, we used the ordinary least squares (OLS) regression model. The model allowed us to observe how the control variables and the predictor variables, which include frequency of use, online relationship size, and Internet proficiency, affect the outcome variable (subjective well-being). We conducted robustness tests by using a substitute outcome variable and an alternative model. We also employed heterogeneity analysis to investigate the role of age differences in the influence of Internet use on subjective well-being. A critical value of *p* < 0.05 was considered to be statistically significant.

## 4. Results

### 4.1. Demographic Characteristics of Study Participants

Table 1 presents the characteristics of the sample. It can be seen that 50.3% of the respondents were women (N = 1046) and that 49.7% were men, with 76.1% of them married. The average age was 41.9 (SD = 14.1), and the average respondent had received approximately 14 years of education (SD = 2.5). The average score for the frequency of Internet usage is 4.1, indicating that the respondents surfed the Internet several times a week or every day. They would connect with three individuals online every day on average (SD = 1.2), and they would organize one virtual community (SD = 3). The average SWB score is 3.9, indicating that the respondents were happy or very happy.

### 4.2. Regression Model Tests

As shown in Table 2, Model 1 explores the control variables that are related to individual subjective well-being. Age is associated with strongly negative effects (B = −0.060, SE = 0.008, *p* < 0.001). It also has a non-linear negative effect on subjective well-being. Education and income have a weak positive effect—respondents with higher education levels and higher annual incomes were happier (B = 0.012, SE = 0.007, *p* < 0.1; B = 0.008, SE = 0.005, *p* < 0.1). Being married is associated with significantly higher subjective well-being (B = 0.418, SE = 0.048, *p* < 0.001). The differences between the SWB levels of respondents from different sexes are not significant.

Model 2, Model 3, and Model 4 yield regression coefficients for Internet usage and subjective well-being. A one-unit increase in Internet use is associated with an increase in the probability of being happy of 0.05 (B = 0.055, SE = 0.017, *p* < 0.01). In the explanatory Model 2, that probability increases by 0.005 relative to the model that only includes the control variables. Model 3 shows that there is a significant association between the online relationship size and subjective well-being (B = 0.039, SE = 0.015, *p* < 0.01). Internet proficiency also benefits subjective well-being (B = 0.016, SE = 0.006, *p* < 0.01). Model 4 points to positive associations between three aspects of Internet usage and subjective well-being (B = 0.044, SE = 0.018, *p* < 0.05; B = 0.032, SE = 0.015, *p* < 0.05; B = 0.016, SE = 0.006, *p* < 0.01). The explained variance increases to 0.056. Therefore, H1, H2, and H3 are supported.

### 4.3. Robustness Tests

We used life satisfaction as a substitute variable. The second column of Table 3 displays the OLS regression of Internet usage on life satisfaction. We also used the ordered probit model to test for robustness. The third column of Table 3 displays the effect of Internet usage on subjective well-being. The two columns exhibit similar regression results and consistent sign directions. Therefore, this study is reliable and robust. The results indicate that the first research question should be answered in the affirmative.

### 4.4. Heterogeneity Analysis

The regression models in Table 2 show that age squared is negatively correlated with subjective well-being and that Internet usage may differently affect different age groups. We divided the sample into three age categories. In line with the definition that is employed by the United Nations and in previous studies [71], we assumed that individuals below the age of 40 are young, that individuals between the ages of 41 and 59 are in middle age, and that individuals over the age of 60 are older adults. Table 4 displays the OLS regression results and the age groups. Frequency of Internet use and the online relationship size have a significant positive relationship with subjective well-being among middle-aged respondents but are not linked to subjective well-being among the young and among older adults (B = 0.063, SE = 0.026, *p* < 0.05; B = 0.043, SE = 0.025, *p* < 0.1). Internet proficiency has a significant relationship with subjective well-being among young and older adults (B = 0.015, SE = 0.007, *p* < 0.05; B = 0.020, SE = 0.012, *p* < 0.10). Therefore, the effect of Internet use on the subjective well-being of individuals of different ages is heterogeneous. The results indicate that the second research question ought to be answered in the affirmative.

## 5. Discussion

### 5.1. Explanations of the Findings

This paper explored the association between Internet use behaviors and subjective well-being by using data from the 2017 CGSS. The findings from the regression model demonstrate that Internet use has a significant positive association with subjective well-being. The results of the robustness test also show that Internet use can promote subjective well-being levels. The heterogeneity tests revealed differences between the subjective well-being that results from the adoption of various Internet use behaviors by users from different age groups.

The results support our prediction that the frequency of Internet use would be positively associated with subjective well-being. This result is inconsistent with the findings of Zeng [72], which indicate that there is no relationship between the two variables. A possible explanation is that Zeng drew on nationwide data from 2015. It is possible that an increase in the frequency of Internet use does not result from repetitive behaviors but from the emergence of interesting and useful activities and new applications that enhance subjective well-being.

The results also demonstrate that online relationship size and Internet proficiency are positively associated with subjective well-being. The number of friends and the level of social interaction have been recognized as two essential elements for social support seeking and provision [73,74]. Previous studies have focused on the relationship between social support, psychological health, and subjective well-being [5,75,76]. The findings are in accord with the results from the studies that indicate that Internet usage results in enhanced socialization and more interpersonal conversations, thus improving social relations and perceived quality of life [77].

Our study demonstrated that organizing interactions in online communities, including in small groups, is more strongly linked to increased subjective well-being than the number of online relationships. As noted in previous studies, group communication favors micro-coordination and small group chats that run in parallel to large ones may serve as backchannels [78]. Additionally, our findings are inconsistent with those of Keipi [36], who, drawing on a survey of Finnish youths, argued that placing trust in weak social ties to individuals whom one has met online is positively associated with subjective well-being. This inconsistency may be due to the differences between samples—the examination of Internet proficiency in this paper is based on pieces of social software such as WeChat and QQ. In addition, values differ across countries [79]. The Finnish emphasize self-expression, while the Chinese emphasize survival-related values that entail communicating and negotiating within groups.

Furthermore, we observed meaningful differences between age groups. Previous research has suggested that hope is an important factor for subjective well-being in groups of teenage friends—individuals from the same friendship group exhibit similar levels of hope [80]. The present study indicates that the association between Internet proficiency and subjective well-being is more obvious among the young. The online relationship size was weakly but positively linked to the subjective well-being of the middle-aged respondents. This unanticipated finding may be explained by the tendency of the popular media, as well as of formal institutions, to present the young as tech-savvy “digital natives” [81]. The young like to participate in virtual communities and explore more of their functions than middle-aged individuals. Moreover, the young often organize or coordinate virtual communications, for example by providing practical guidance to guests via WeChat groups and by collecting feedback from purchasers via QQ groups. Their centrality to online interactions and the recognition that they receive as a result improves their subjective well-being.

The results indicate that a higher frequency of Internet use is not linked to improved subjective well-being among older individuals, possibly because users need certain skills to make full use of the available Internet applications. We also demonstrated that gains in well-being are more likely to be maximized when the Internet is prudently and skillfully used [82]. Older individuals who organize virtual communities actively enjoy higher levels of subjective well-being. Loneliness impacts the quality of life among older adults, a problem that is becoming more acute due to the aging of the world population and COVID-19 measures [83]. The present study indicates that active interaction, especially if one is at the center of a network, is linked to improved subjective well-being. One possible explanation is that relatives and friends monitor the living conditions and the psychological health of older individuals, which decreases negative emotions and promotes a sense of well-being.

The practical implications of these results are significant. First, policymakers should encourage the public to use the Internet in order to improve their subjective well-being. Individuals can enjoy life by making full use of the functionalities of the Internet, such as information provision, social communication, and entertainment. Second, Internet services should be differentiated between age groups. Our study has shown that middle-aged individuals benefit from more frequent Internet use and larger-scale networks; young and older adults benefit from organizing communication in groups. Therefore, policymakers and software companies can design Internet services and applications with richer features to reflect the usage habits of middle-aged individuals. Older individuals can improve their media literacy to adapt to the Internet. Third, more attention should be paid to Internet use among older adults as populations age. Social support, in particular, should not be neglected. For example, older individuals should be encouraged to set up interactive online groups for their children, relatives, and friends and to actively share their knowledge and views. By networking, they can obtain emotional and instrumental support as well as avoid loneliness.

It should be noted that the present study has several limitations. Follow-up studies are needed to identify the impact of social support from online networks. In addition, a longitudinal and experimental design would be necessary to confidently infer and to identify the short- and long-term consequences of Internet use.

### 5.2. Limitations and Future Directions

It should be noted that there are some limitations to this study. First, the data itself has limitations. The data used in this study is not the latest, and with the rapid growth of the number of Internet users in China [22,66], future research can be analyzed based on the latest data to consider the impact of Internet usage behaviors on subjective well-being. In addition, because this study used second-hand data [84], the measurement items were limited. Hence, we only take factors of frequency of Internet use, online relationship size, and Internet proficiency into the study, and future studies can include other influencing factors. Previous studies have found that social support [75,76] and social networking [47] are also linked to online networks, and researchers can incorporate these variables in conjunction with subjective well-being for further consideration.

Second, the behaviors of excessive internet use and the type of content have not been explored. This paper mainly discusses the association between the three aspects of internet usage behaviors and subjective well-being and does not focus on excessive Internet use. However, long-term internet usage can also increase anxiety [43,44]. Researchers can also investigate the impact of excessive internet usage behaviors on subjective well-being. In terms of the content of internet use, the subjective well-being of individuals who are excessively exposed to negative content is relatively low [37]. People in future research directions can appropriately consider different types of internet content to discuss the impact of various internet behaviors on subjective well-being.

Finally, further exploration of the longitudinal study of internet use behaviors and subjective well-being is worthy of consideration. Based on the cross-sectional data adopted, this paper discusses the association between Internet use behaviors and subjective well-being. However, the rapid development of the Internet, even mobile Internet, and its long-term effects on individuals’ psychological health is a topic that deserves more consideration in today’s rapidly developing society [21,52,83]. Therefore, A longitudinal and experimental design would be needed to confidently infer causality and the short- and long-term consequences of Internet use.

## 6. Conclusions

The results from the analysis of a nationally representative Chinese sample confirm that there is a significant and stable relationship between the aspects of Internet use, including the frequency of Internet use, online relationship size, Internet proficiency, and subjective well-being.

Moreover, this study creatively identifies that the association between Internet use and subjective well-being significantly varies between age groups. The subjective well-being of middle-aged individuals benefits from more frequent Internet use and larger-scale networks. Young and older adults benefit from organizing communication in groups; that is to say, Internet proficiency is clearly associated with young and old people’s subjective well-being. In addition, for the elderly, the frequency of Internet use is positively correlated with subjective well-being. The findings of this study help extend the literature and provide targeted suggestions for improving the subjective well-being of different age groups in Internet use.

## Figures and Tables

**Table 1 ijerph-20-02897-t001:** Demographics and characteristics of participants (N = 2079).

Variables	n (%)	Mean (SD)
Sex		
Female	1046 (50.30%)	
Male	1033 (49.70%)	
Marital status		
Unmarried	497 (23.90%)	
Married	1582 (76.10%)	
Age		41.9 (±14.1)
Education level		13.90 (±2.55)
Personal annual income		8.86 (±3.86)
Use frequency		4.09 (±1.08)
Online relationship size		3.12 (±1.20)
Internet proficiency		0.98 (±14.1)
Subjective well-being		3.90 (±14.1)

**Table 2 ijerph-20-02897-t002:** Regression analysis showing the relationship between Internet use variables and subjective well-being (N = 2079).

Variables	Model 1	Model 2	Model 3	Model 4
	B (SE)	B (SE)	B (SE)	B (SE)
Sex	−0.028	−0.030	−0.031	−0.032
	(0.034)	(0.034)	(0.034)	(0.034)
Age	−0.060 ***	−0.058 ***	−0.057 ***	−0.056 ***
	(0.008)	(0.008)	(0.008)	(0.008)
Age^2^	0.001 ***	0.001 ***	0.001 ***	0.001 ***
	(0.000)	(0.000)	(0.000)	(0.000)
Education level	0.012 +	0.010	0.010	0.008
	(0.007)	(0.007)	(0.007)	(0.007)
Personal annual income	0.008 +	0.007	0.005	0.004
	(0.005)	(0.005)	(0.005)	(0.005)
Marital status	0.418 ***	0.411 ***	0.416 ***	0.411 ***
	(0.048)	(0.048)	(0.048)	(0.048)
Use frequency		0.055 **		0.044 *
		(0.017)		(0.018)
Online relationship size			0.039 **	0.032 *
			(0.015)	(0.015)
Internet proficiency			0.016 **	0.016 **
			(0.006)	(0.006)
Constant	4.627 ***	4.366 ***	4.455 ***	4.269 ***
	(0.184)	(0.201)	(0.189)	(0.203)

Note: Unstandardized coefficients. + *p* < 0.1; * *p* < 0.05; ** *p* < 0.01; *** *p* < 0.001.

**Table 3 ijerph-20-02897-t003:** Regression results for the robustness of the substituted explained variable and analysis model (N = 2079).

Variables	Subjective Well-Being OLS	Life Satisfaction OLS	Subjective Well-Being Ordered Probit
	B (SE)	B (SE)	B (SE)
Use frequency	0.044 *	0.057 *	0.059 *
	(0.018)	(0.023)	(0.026)
Online relationship size	0.032 *	0.043 *	0.049 *
	(0.015)	(0.020)	(0.022)
Internet proficiency	0.016 **	0.013 +	0.024 **
	(0.006)	(0.007)	(0.009)
Sex	−0.032	−0.057	−0.027
	(0.034)	(0.045)	(0.050)
Age	−0.056 ***	−0.049 ***	−0.081 ***
	(0.008)	(0.011)	(0.012)
Age^2^	0.001 ***	0.001 ***	0.001 ***
	(0.000)	(0.000)	(0.000)
Education level	0.008	0.020*	0.011
	(0.007)	(0.009)	(0.010)
Personal annual income	0.004	0.006	0.007
	(0.005)	(0.006)	(0.007)
Marital status	0.411 ***	0.371 ***	0.574 ***
	(0.048)	(0.063)	(0.070)
Constant	4.269 ***	4.782 ***	
	(0.203)	(0.267)	
cut 1			−2.947 ***
			(0.309)
cut 2			−2.151 ***
			(0.301)
cut 3			−1.435 ***
			(0.299)
cut 4			0.385
			(0.298)
N	2079	2079	2079
R^2^/pseudo R^2^	0.056	0.041	0.026

Note: unstandardized coefficients. + *p* < 0.1; * *p* < 0.05; ** *p* < 0.01; *** *p* < 0.001.

**Table 4 ijerph-20-02897-t004:** Regression results of age heterogeneity of Internet use associated with subjective well-being (N = 2079).

Variables	Young Group	Middle-Aged Group	The Older Group
	B (SE)	B (SE)	B (SE)
Sex	−0.022	−0.046	−0.015
	(0.050)	(0.057)	(0.093)
Education level	0.017 +	0.007	−0.011
	(0.010)	(0.012)	(0.018)
Personal annual income	−0.004	0.006	0.024
	(0.006)	(0.008)	(0.022)
Marital status	0.190 ***	0.590 ***	0.366 **
	(0.052)	(0.099)	(0.116)
Use frequency	0.054	0.063 *	−0.008
	(0.034)	(0.026)	(0.037)
Online relationship size	0.025	0.043 +	0.019
	(0.022)	(0.025)	(0.036)
Internet proficiency	0.015 *	0.024	0.020 +
	(0.007)	(0.021)	(0.012)
Constant	3.222 ***	2.784 ***	3.615 ***
	(0.207)	(0.213)	(0.317)
*N*	1020	792	267
R^2^/pseudo R^2^	0.026	0.068	0.053

Note: unstandardized coefficients. + *p* < 0.1; * *p* < 0.05; ** *p* < 0.01; *** *p* < 0.001.

## Data Availability

This paper uses data from the Chinese General Social Survey (CGSS), a nationally representative longitudinal survey committed by the National Survey Research Center at Renmin University of China (NSRC), sponsored by the RUC 985 Grant and RUC Scientific Research Grant. The datasets analyzed during the current study are available from the corresponding author upon reasonable request.

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
