# Peer review of "Does Internet Use Promote Subjective Well-Being? Evidence from the Different Age Groups Based on CGSS 2017 Data"

_ijerph, 2023, doi:10.3390/ijerph20042897_

Round 1
Reviewer 1 Report
This is a well-performed study focusing on an intriguing topic. The authors adopted a national representative dataset to empirically examine the relationship between Internet usage and subjective well-being, which adds new knowledge to this classic but inconclusive issue. This manuscript possesses the following merits - first, it decomposes Internet usage into three dimensions, enriching the current understanding of media use (as a multidimensional and composite concept). Second, age, which has been somewhat overlooked in previous studies of its kind, receives an in-depth analysis in this work. The heterogeneous results across different age groups demonstrate that Internet usage's effects on subjective well-being should be investigated from a nuanced perspective. Third, the authors employed different methods to perform the robustness check, including utilizing a proxy outcome variable and an alternative model. All the efforts significantly strengthen the reliability and validity of the results.
However, in addition to all the strengths, some weaknesses should be pointed out to improve the quality of this study. Two major concerns are listed as follows.
1 - Although the authors summarized three dimensions of Internet usage (i.e., frequency of use, online relationship size, and online relationship centrality), the latter two dimensions seem like the outcomes of the first dimension. For instance, as indicated in line 131 - "online communication expands relationship networks for communication and further enhances subjective well-being." Thus, a reasonable conjecture could be that frequent Internet use leads to online relationship size extension, further contributing to subjective well-being. Similarly, an experienced Internet user may have a greater probability of obtaining prestige in the relationship network (i.e., a high network centrality), which benefits subjective well-being. Therefore, why the authors adopted an isolated perspective to explore the three dimensions instead of building mediation processes? I suggest the authors develop their hypotheses from another angle. For example, Internet use comprises three aspects (i.e., frequency, duration, and proficiency), in which frequency is straightforward and corresponds to H1. Duration, on the other hand, helps to expand the network size, which is associated with the second hypothesis. In a similar vein, proficiency denotes organizational ability, such as building WeChat or QQ groups, which is closely related to the third hypothesis. However, this is a tentative suggestion. The authors may find a more appropriate angle to elaborate on the three dimensions. In summary, the core principle of revision is to demonstrate the three dimensions as separate and independent rather than interwoven ones.
2 - In the third paragraph of the discussion section, the authors mentioned the term "social support." However, I didn't find a solid argument to bolster the claim that online relationship size and online relationship centrality could boost social support. If the authors want to use social support as a possible explanation, then a more in-depth discussion is necessitated.
Some minor suggestions are listed as follows.
1 - All tables need to be polished, especially the table headings and notes. For instance, what are the values in parentheses in Table 2?
2 - Please avoid using informal terms in the main text. For example, the "age2 variable" in line 214 is inappropriate and should be replaced by "the square of age."
3 - You don't need to write the expression of the OLS model out in the main text. Otherwise, for the ordered probit model, you have to present the expression as well.
4 - I recommend the authors conduct a round of thorough proofreading to address all the grammatical errors. For example, the correct format of the sentences at the beginning should be as follows.
"With the rapid development of mobile Internet technology, the Internet plays an indispensable role in everyday life. People [in which country or within what geographical scope?] spend an average of 7 hours per day using the Internet, with an average of 145 minutes per day using social media."
Reviewer 2 Report
The article addresses a highly topical issue. The article is written in a hijacked structure. A relatively large group of respondents took part in the research. The research process was carried out correctly using statistical methods. The literature review and research methodology were solidly described. I missed a clear indication of the definition of well-being used in the research.
The research dates from 2017. They were conducted according to the principles of research methodology. The question arises whether, 5 years after the research was conducted, the results are still valid. - arguments should be provided for your defence.
The issue related to the dangers of excessive internet use was not addressed. I feel unsatisfied with the presentation of the need for further research, and the definition of the scope of such research. The section on limitations also needs to be expanded.
Other specific comments:
-changing the title to an indicative form
- complement the conclusions with a clear answer to the research question
- improve table descriptions - technical suggestion
